# Simplifying Multi-Task Architectures Through Task-Specific Normalization

## Abstract

Multi-task learning (MTL) aims to leverage shared knowledge across tasks to improve generalization and parameter efficiency, yet balancing resources and mitigating interference remain open challenges. Architectural solutions often introduce elaborate task-specific modules or routing schemes, increasing complexity and overhead. In this work, we show that normalization layers alone are sufficient to address many of these challenges. Simply replacing shared normalization with task-specific variants already yields competitive performance, questioning the need for complex designs. Building on this insight, we propose Task-Specific Sigmoid Batch Normalization (TS$\sigma$BN), a lightweight mechanism that enables tasks to softly allocate network capacity while fully sharing feature extractors. TS$\sigma$BN improves stability across CNNs and Transformers, matching or exceeding performance on NYUv2, Cityscapes, CelebA, and PascalContext, while remaining highly parameter-efficient. Moreover, its learned gates provide a natural framework for analyzing MTL dynamics, offering interpretable insights into capacity allocation, filter specialization, and task relationships. Our findings suggest that complex MTL architectures may be unnecessary and that task-specific normalization offers a simple, interpretable, and efficient alternative.

## 1 Introduction

Multi-task learning (MTL) trains a single model to solve multiple tasks jointly, leveraging shared representations to improve generalization and computational efficiency. Despite many successes, MTL remains difficult to understand and control. Core challenges include task interference, where competing gradients from divergent task requirements disrupt joint training (Zhang et al., 2022); capacity allocation, where shared and task-specific resources must be balanced to avoid dominance (Maziarz et al., 2019; Newell et al., 2019); and task similarity, where the degree of relatedness determines how tasks should interact (Standley et al., 2020). Existing approaches typically address only one of these issues. Optimization-based methods focus on mitigating interference by reweighting losses or modifying gradients (Yu et al., 2020; Navon et al., 2022). Soft-sharing architectures attempt to disentangle capacity by adding task-specific modules on top of a shared backbone, but in doing so often introduce significant design complexity in deciding how modules should interact (Misra et al., 2016; Liu et al., 2019). Neural architecture search methods learn to partition networks based on data-driven estimates of task-relatedness (Guo et al., 2020; Sun et al., 2020).

In this work, we argue that normalization layers and in particular batch normalization (BN) (Ioffe, 2015) are a sufficient and highly effective solution for all the aforementioned challenges in MTL. Our motivation stems from the following observations:

First, while neural networks are heavily over-parameterized, existing approaches struggle to resolve tasks conflicts (Shi et al., 2023), indicating a failure to utilize the available network capacity optimally. Second, BN has proven to be highly expressive - not only does it stabilize and accelerate training (Santurkar et al., 2018; Bjorck et al., 2018), but it also demonstrates remarkable standalone performance when used on random feature extractors (Rosenfeld & Tsotsos, 2019; Frankle et al., 2021) and its ability to leverage features not explicitly optimized for a specific task (Zhao et al., 2024). Third, BN can learn to ignore unimportant features (Frankle et al., 2021) or be explicitly regularized to produce structured sparsity (Liu et al., 2017; Suteu & Guo, 2022). This can be leveraged for MTL when unrelated tasks cannot fully share all features without interference and require disentanglement. Fourth, normalization layers are extremely parameter-efficient, taking up typically less than 0.5% of a

model's size. This makes them particularly suitable as lightweight universal adapters for applications where models need to scale to multiple tasks (Rebuffi et al., 2017; Bilen & Vedaldi, 2017).

Lastly, while conditional BN layers have been explored in settings with domain shift (Wallingford et al., 2022; Xie et al., 2023; Chang et al., 2019; Deng et al., 2023), these methods focus on the issue of mismatched normalization statistics and use task-specific BN as a domain-alignment tool. Our focus is different: we study single-domain MTL, where all tasks share the same input distribution and normalization does not become a failure mode. In this setting, we show that task-specific BN can provide a simple way to modulate representations via their affine parameters - turning it from a normalization module into a lightweight mechanism for capacity allocation and interference reduction. The extension of BN as the sole mechanism for modulation and interpretability rather than domain alignment remains largely unexplored.

Motivated by these observations, we propose a minimalist soft-sharing approach to MTL, where feature extractors are fully shared and only normalization layers are task-specific. Unlike prior soft-sharing architectures that add complex modules or routing schemes, our design isolates normalization as the sole mechanism for balancing tasks. Building on $\sigma$BN (Suteu & Guo, 2022), we introduce lightweight task-specific gates that modulate feature usage with negligible overhead, making the approach broadly compatible, easy to implement, and resilient to task imbalance. Beyond performance and efficiency, the learned $\sigma$BN parameters naturally form a task-filter importance matrix, enabling a structured analysis of capacity allocation, filter specialization, and task relationships, providing an interpretable view of MTL that is largely absent in prior work.

**Contributions:**

- A minimal MTL baseline. We show that simply replacing shared normalization with task-specific BatchNorm (TSBN) already delivers competitive performance out-of-the-box, questioning the necessity of elaborate task-specific modules or routing schemes.

- An extended design with sigmoid normalization. We introduce TS$\sigma$BN which improves stability and scale across CNNs and transformers. This variant achieves superior performance on nearly all benchmarks while remaining parameter-efficient.

- An interpretable analysis framework. The use of $\sigma$BN further provides a natural lens for analyzing MTL dynamics. By interpreting learned feature importances, we obtain structured insights into capacity allocation, filter specialization, and task relationships.

## 2 RELATED WORK

**Soft parameter sharing** methods tackle MTL interference architecturally by introducing task-specific modules to a shared backbone. Design options include replicating backbones (Misra et al., 2016; Ruder et al., 2019), adding attention mechanisms (Liu et al., 2019; Maninis et al., 2019), low-rank adaptation modules (Liu et al., 2022b; Agiza et al., 2024) or allowing cross-talk at a decoder level (Xu et al., 2018; Vandenhende et al., 2020b). However, these methods rely on task-specific feature extractors to avoid negative transfer at the cost of forgoing the multi-task inductive bias. Furthermore, adding task-specific capacity scales poorly with many tasks (Strezoski et al., 2019), and requires extensive code modifications that hinder adaptation to new architectures. Although BatchNorm is present in many of these systems, it is embedded in larger task-specific designs. In contrast, our method isolates BatchNorm as the sole soft-sharing mechanism, showing that it is a sufficient solution for competitive MTL while challenging unnecessary complexity.

**Neural Architecture Search (NAS)** methods reduce task interference by choosing which parameters to share among tasks as hard-partitioned sub-networks. Some approaches use probabilistic sampling (Sun et al., 2020; Bragman et al., 2019; Maziarz et al., 2019; Newell et al., 2019) or explicit branching/grouping strategies based on task affinities (Vandenhende et al., 2020a; Guo et al., 2020; Bruggemann et al., 2020; Standley et al., 2020; Fifty et al., 2021). Others use hypernetworks (Raychaudhuri et al., 2022; Aich et al., 2023) which learn to generate MTL architectures conditioned on user preferences. While our method also models task relationships and capacity allocation, it does so without architecture search, relying solely on static modulation via normalization layers.

**Mixture-of-Experts (MoE)** methods address task interference by dynamically routing inputs to specialized experts, enabling flexible capacity allocation among tasks (Ma et al., 2018; Hazimeh et al., 2021; Tang et al., 2020). More recent work extends MoE designs to large-scale transformer architectures for vision and language tasks (Fan et al., 2022; Chen et al., 2023; Ye & Xu, 2023; Yang et al., 2024). Although effective, these methods rely on dynamic, per-sample routing that increases architectural and training complexity. In contrast, our approach provides a static and lightweight form of soft partitioning, achieving similar benefits with minimal changes to the wrapped backbone.

**Parameter-efficient fine-tuning (PEFT)** is a popular approach for adapting large pre-trained models without updating the full backbone. Single-task PEFT methods such as Adapters (He et al., 2021), BitFit (Zaken et al., 2022), VPT (Jia et al., 2022), Compacter (Karimi Mahabadi et al., 2021), and LoRA-style updates add small task-specific modules or low-rank layers while keeping most weights frozen. Extending these ideas to MTL requires managing several task-specific adapters at once. Recent PEFT-MTL methods address this by generating adapter weights through hypernetworks or decompositions, as in HyperFormer (Mahabadi et al., 2021), Polyhistor (Liu et al., 2022b), and MTLoRA (Agiza et al., 2024). However, these methods still rely on additional task-specific capacity, which parallels traditional soft-parameter sharing and scales poorly with the number of tasks. In contrast, we modulate the shared capacity directly through BN, without adding new feature extractors.

**Domain-specific normalization** has become a common technique in settings with domain shift, where shared BatchNorm fails because domains have different input distributions. In these cases, separate BN statistics or layers are required to maintain stable normalization (Li et al., 2016; Zajac et al., 2019; Chang et al., 2019). The same motivation appears in several areas: In meta-learning, TaskNorm (Bronskill et al., 2020) adapt BN statistics per episode to handle changes in input distribution. In continual learning, CLBN (Xie et al., 2023) store task-specific BN parameters to avoid catastrophic forgetting from normalization drift. In conditional or multi-modal models, BN and LayerNorm is adjusted to match modality-specific statistics (Michalski et al., 2019; Zhao et al., 2024). In multi-domain MTL (Bilen & Vedaldi, 2017; Mudrakarta et al., 2019; Wallingford et al., 2022; Deng et al., 2023), task-specific BN is used as an adapter for tasks from different domains. In contrast, our work targets single-domain MTL, where all tasks share the same input and normalization does not fail. In this case, task-specific BN is not needed for statistical correction. Instead, we focus on its affine parameters as a basis for task-specific feature modulation, and extend this idea with a reparameterization and optimization scheme tailored to reduce interference and allocate capacity.

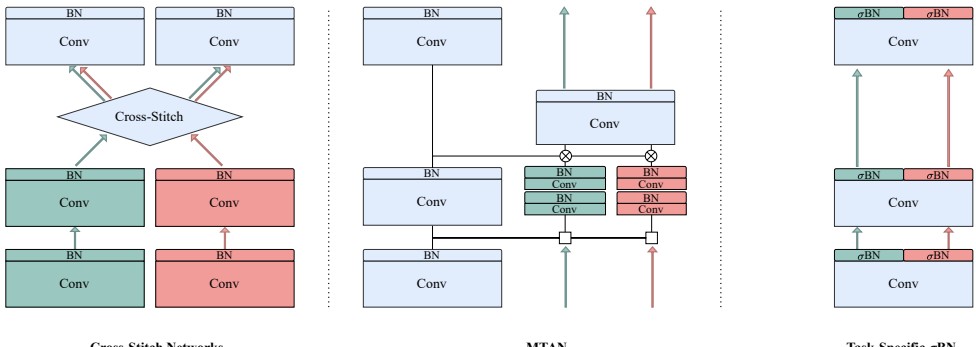

Figure 1: Illustration of soft parameter sharing architectures in a two-task setting. Cross-Stitch Networks (Misra et al., 2016) and MTAN (Liu et al., 2019) incorporate additional feature extractors, which lead to scalability challenges as the number of tasks increases. Task-Specific $\sigma$BN Networks introduce only task-specific normalization layers, offering a highly parameter-efficient solution.

## 3 BATCHNORM AND $\sigma$BATCHNORM

Batch normalization is a cornerstone for deep CNNs due to its versatility, efficiency, and wide-ranging benefits, including improved training stability for faster convergence (Santurkar et al., 2018; Bjorck et al., 2018), regularization effects (Luo et al., 2019), and the orthogonalization of representations (Daneshmand et al., 2021). BN operates in two key steps - normalization and affine transformation:

$$BN(x; \gamma, \beta) = \gamma \hat{x} + \beta, \qquad \hat{x} = \frac{x - \mu_B}{\sqrt{\sigma_B^2 + \epsilon}} \tag{1}$$

The normalization step standardizes input activations using the mini-batch mean $\mu_B$ and variance $\sigma_B^2$, while the affine transformation applies channel-specific learnable parameters, $\gamma$ and $\beta$, to re-scale and shift the normalized activations. During inference, BN relies on population statistics collected during training via running estimates. When the test distribution differs from the training set, these statistics can become mismatched and significantly degrade model performance (Summers & Dinneen, 2020). Because of this, many BN variants aim to improve the normalization step itself by adjusting $\mu$ and $\sigma$ to handle distribution changes, domain shift, meta-learning episodes, or multi-modal inputs. For a survey on normalization approaches we refer to Huang et al. (2023).

In single-domain MTL, all tasks share the same input distribution, so the normalization component of BN does not need adjustment. Instead, we focus on the affine transformation post-normalization. These parameters represent only a small fraction of the network, yet they have substantial expressive power, as shown by studies demonstrating high performance when training BN alone (Frankle et al., 2021). In this work, we build on a variation of BN originally introduced to determine feature importance in structured pruning, Sigmoid Batch Normalization (Suteu & Guo, 2022) replaces the affine transformation with a single bounded scaler:

$$\sigma BN(x; \gamma) = \sigma(\gamma)\hat{x}, \qquad \sigma(\gamma) = \frac{1}{1 + e^{-\gamma}} \tag{2}$$

Using a single bounded scaler per feature has little impact on performance, but enables targeted regularization and improves interpretability. These properties make $\sigma$BN especially attractive for multi-task learning, where understanding how tasks share limited capacity is critical. In this setting, $\sigma(\gamma)$ acts as a static soft gate that can down-weight or disable features. This implicit static gating contrasts with soft-sharing models, which explicitly partition capacity, and MoE methods, which route features dynamically through task-specific gates. Furthermore, this formulation can be extended to other normalization layers (Ba et al., 2016), as we show in experiments on transformers. Using $\sigma$BN as the only task-specific components, we create a parameter-efficient framework that sustains performance while providing tools to analyze and influence capacity allocation and task relationships.

## 4 TASK-SPECIFIC $\sigma$BATCHNORM NETWORKS

TS$\sigma$BN networks are constructed by replacing every shared Batch Normalization layer with task-specific $\sigma$BN layers, as illustrated in Figure 1. This design allows tasks to normalize and modulate the outputs of shared convolutional layers:

$$TS\sigma BN(x; \gamma_t) = \sigma(\gamma_t)\hat{x}, \qquad \hat{x} = \frac{x - \mu_{B,t}}{\sqrt{(\sigma_{B,t})^2 + \epsilon}} \tag{3}$$

enabling better disentanglement of representations and reduced task interference. Unlike prior methods introducing additional task-specific capacity, TS$\sigma$BN keeps all convolutions shared, preserving the multi-task learning inductive bias toward generalizable representations. While domain-specific BN has been used reactively in domain adaptation (Chang et al., 2019) to handle distribution shifts, our work is the first to use it proactively as a standalone mechanism in single-input scenarios.

**Task interference**. Conflicting gradient updates between tasks is a central challenge in MTL, often measured by negative cosine similarity (Zhao et al., 2018; Yu et al., 2020; Shi et al., 2023). Figure 2 (left) shows the gradient similarity distribution for shared convolutional parameters: in hard parameter sharing, the distribution is nearly uniform, meaning roughly half of all updates conflict. MTAN

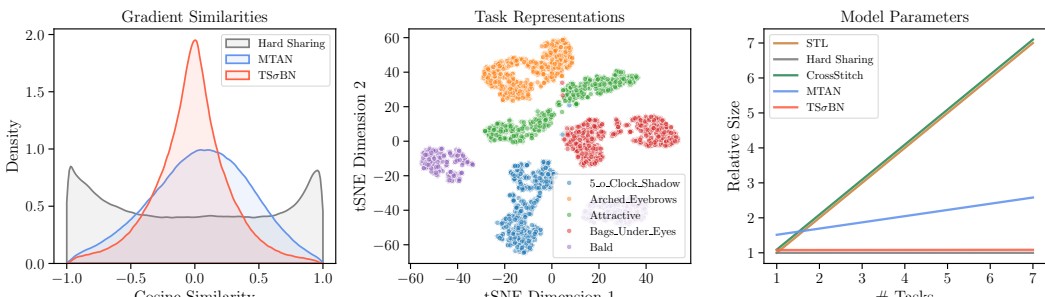

Figure 2: Left: Distribution of cosine similarities between the gradients of NYUv2 tasks over the shared convolutions in the early stages of training. Middle: t-SNE visualization of the encoder representations for the first five CelebA tasks. Right: Encoder parameter count for various numbers of tasks relative to a ResNet50 backbone. Overall, TS$\sigma$BN has a greater concentration of orthogonal gradients, produces well-separated task representations and has a negligible parameter growth.

(Liu et al., 2019) partially alleviates this issue by introducing task-specific convolutions. In contrast, TS$\sigma$BN yields a sharp, zero-centered distribution with low variance, indicating gradients are mostly orthogonal. This mirrors optimization-based methods that explicitly enforce orthogonality (Yu et al., 2020; Suteu & Guo, 2019), yet TS$\sigma$BN achieves it through a lightweight architectural change. Figure 2 (middle) further supports this: on CelebA, task representations form well-separated clusters, illustrating reduced interference. A full analysis across all tasks is provided in Appendix A.

**Parameter Efficiency.** Task-Specific $\sigma$BN is highly parameter efficient since it does not introduce additional feature extractors like related soft parameter sharing architectures. At the extreme end, such as Single Task Learning or Cross-Stitch networks, the entire backbone is duplicated for each new task. TS$\sigma$BN on the other hand duplicates only $\sigma$BN layers, whose parameters comprise a fraction of the total model size. Figure 2 (right) shows how different approaches scale with additional tasks. TS$\sigma$BN adds an insignificant amount of new parameters, allowing it to scale to any number of tasks.

**Discriminative Learning Rates**. We increase the learning rate of $\sigma$BN parameters by a fixed multiple ($\alpha_{\sigma BN} = 10^2$) relative to other parameters, allowing them to allocate filters before these undergo significant updates. This accelerates specialization and ensures capacity allocation occurs early in training. A further advantage of $\sigma$BN is its robustness to high learning rates: the sigmoid dampens gradients, making training stable across scales, whereas vanilla BN is more sensitive and requires careful tuning. The approach parallels transfer learning, where deeper layers are updated more aggressively to drive adaptation (Howard & Ruder, 2018; Vlaar & Leimkuhler, 2022). We provide ablations on how higher learning rates improve performance and filter allocation.

## 5 MTL ANALYSIS WITH TS$\sigma$BN

A key advantage of the TS$\sigma$BN design is the ability to quantify filter allocation through task-filter importance matrices. Since each $\sigma$BN layer introduces a dedicated scaling parameter $\gamma_{t,i}$ per task and filter, we construct a task-filter importance matrix $I \in \mathbb{R}^{T \times F}$, where each entry $I_{t,i}$ captures the importance task $t$ assigns to filter $i$. Applying the sigmoid function to the raw scaling parameters $I_{t,i} = \sigma(\gamma_{t,i})$ ensures that values remain within $[0, 1]$, facilitating interpretability and comparability across tasks, layers, and models. Using this representation, TS$\sigma$BN enables a principled analysis of MTL dynamics, including capacity allocation, task relationships, and filter specialization.

### 5.1 CAPACITY ALLOCATION

One of the central challenges in multi-task learning is understanding how model capacity is allocated among competing tasks. The TS$\sigma$BN task-filter importance matrix $I$ can directly quantify the total capacity of a task $t$ as the normalized sum of the importances it assigns to filters $C_t = \frac{1}{F} \sum_{i=1}^{F} \sigma(\gamma_{t,i})$. This measure provides an overall assessment of the resources required for each task; however, it does

not account for task relationships or shared capacity. A task with high absolute capacity does not necessarily imply it monopolizes filters, as it may rely heavily on shared generic filters.

We apply an orthogonal projection-based decomposition to differentiate between task-specific and shared capacity. Given the set of task importance vectors $\{I_1, I_2, ..., I_T\}$, we decompose each task's capacity into an independent component and a shared component. Let $A$ be the matrix formed by stacking all task importance vectors except $I_t$. The projection of $I_t$ onto the subspace spanned by the other tasks is given by the projection matrix $P_A$:

$$P_A I_t = A(A^T A)^{-1} A^T I_t, \tag{4}$$

The shared $\hat{I}_t = P_A I_t$ and independent $I_t^\perp = I_t - \hat{I}_t$ components of $I_t$ can therefore be defined so that $I_t^\perp$ is orthogonal to the subspace spanned by the other task importance vectors.

To derive a capacity decomposition consistent with the original measure, we define the independent and shared capacities as scaled versions of the total capacity:

$$C_t^{indep} = \frac{\|I_t^\perp\|_2}{\|I_t\|_2} C_t, \qquad C_t^{shared} = \frac{\|\hat{I}_t\|_2}{\|I_t\|_2} C_t. \tag{5}$$

Because in this formulation the components are orthogonal, the $L_2$ norm satisfies the Pythagorean theorem, yielding $C_t^2 = (C_t^{shared})^2 + (C_t^{indep})^2$. This guarantees that a task's total capacity is preserved while providing an interpretable split between shared and independent resource usage.

Using our framework, we analyze task capacity allocation after training as shown in Figure 3. For both SegNet and DeepLabV3 architectures, we find that most capacity is shared among tasks without a single task dominating. For a more detailed analysis on the effects of task difficulty and similarity on capacity allocation, we refer to Appendix E. Overall, this view offers interpretability into the interaction between tasks and can be a powerful tool in real-world applications where relationships are not known a priori.

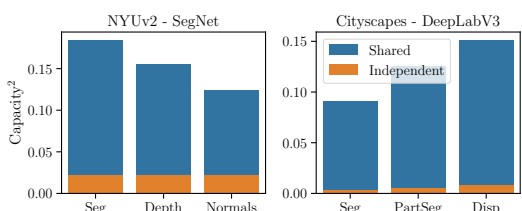

Figure 3: Decomposed task capacity into shared and independent components using the TS$\sigma$BN framework. In all standard scenarios, tasks share most capacity without signs of dominance.

## 5.2 TASK RELATIONSHIPS

A desirable feature for any multi-task learning model is the ability to derive task relationships, as this can help gauge interference between tasks and provide insights into the joint optimization process. To showcase this, we use the CelebA dataset, containing 40 binary facial attribute tasks, allowing us to explore complex task relationships and hierarchies via TS$\sigma$BN. Moreover, because these attributes are semantically interpretable (e.g., "Smiling", "Mouth Slightly Open"), they enable meaningful qualitative assessments of the learned relationships.

To derive task relationships we compute the pairwise cosine similarity between the task importance vectors $I_t \in \mathbb{R}^F$, yielding a $T \times T$ similarity matrix, with values ranging from 0 (orthogonal filter usage) to 1 (indicating identical usage). We use this as the basis for constructing distance matrices to identify task clusters and hierarchical relationships that reflect the model's capacity allocation.

To assess the stability of the task relationships derived from our model, we focus on the consistency of task hierarchies across multiple training runs. Specifically, we evaluate the similarity matrices obtained from seven independently trained models with different intializations. We compute the pairwise Spearman rank correlation between similarity matrices to determine whether the relative task orderings are robust to such variations. Our results show that the task hierarchies are highly stable, with an average Spearman correlation of 0.8 across all model pairs.

We further assess the resulting relationships by aggregating the representative task clusters from the seven runs, via co-occurrence matrices and hierarchical clustering. The identified clusters exhibit

semantic coherence, suggesting a correlation with the spatial proximity of facial attributes. For instance, tasks related to hair characteristics (e.g., Bangs, Blond Hair) form a distinct cluster. In contrast, facial hair attributes (e.g. Goatee, Mustache) are grouped separately. More details about the procedure and resulting task clusters can be found in the Appendix C.

### 5.3 Filter Groups

A different way to analyze multi-task learning is from an individual filter perspective. Using the task-filter matrix, we can gauge each task's reliance on a filter to determine if the resource is specialized or generic. We define a filter as specialized for a particular task if its normalized task-filter importance exceeds a threshold $\tau$. We set $\tau = 0.5$ to signify that the filter predominantly contributes to a single task rather than being shared among multiple tasks. Formally, let $\sigma(\gamma_{t,i})$ denote the importance of filter $i$ for task $t$. A filter $i$ is deemed specialized for task $t'$ if $\sigma(\gamma_{t',i})/\sum_t^T \sigma(\gamma_{t,i}) > \tau$.

We prune the top 200 most important filters per task to test our definitions of specialization and

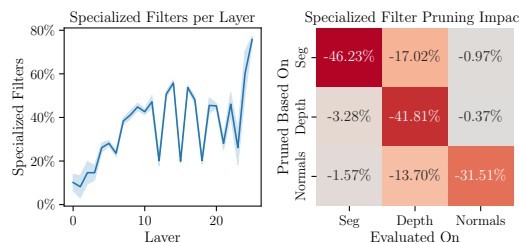

Figure 4: Left: Percentage of specialized filters per layer in a TS$\sigma$BN SegNet. Specialization increases in the latter layers. Right: Performance drop across tasks (columns) after pruning filters based on their primary specialization (rows).

importance. If accurate, removing a task's specialized filters should degrade its performance more than others. Figure 4 (right) confirms this: diagonal elements, representing self-impact, show significantly larger drops than off-diagonals, supporting our hypothesis.

Next, we examine where specialized filters occur across the network. Figure 4 (left) shows the percentage of specialized filters per layer from different runs. Specialization increases with network depth, indicating that early layers are more shared while deeper layers become task-specific. This mirrors findings in single-task learning (Yosinski et al., 2015), where lower layers encode general features, and aligns with branching-based NAS heuristics (Bruggemann et al., 2020; Vandenhende et al., 2020a; Guo et al., 2020), which assign specialized layers to later stages. Our method for quantifying specialization and task similarity offers an alternative perspective for NAS strategies.

## 6 Experiments

We evaluate TS$\sigma$BN across a wide range of MTL settings - covering three CNN (from scratch and pretrained) and two vision transformer architectures over four standard MTL datasets: NYUv2 (Silberman et al., 2012), Cityscapes (Cordts et al., 2016), CelebA (Liu et al., 2015) and PascalContext (Chen et al., 2014). We follow established protocols from prior work (Liu et al., 2019; Ban & Ji, 2024; Lin & Zhang, 2023; Yang et al., 2024; Agiza et al., 2024) for training, evaluation, and metric reporting. TS$\sigma$BN achieves comparable or superior performance to related and state-of-the-art methods while maintaining better resource efficiency. We refer to Appendix F for additional details on TS$\sigma$BN integration, datasets, protocols and baselines.

**Convolutional Neural Networks.** We evaluate TS$\sigma$BN on CNNs in two settings: models trained from scratch and initialized from pretrained backbones. For models trained from scratch, we follow standard protocols on NYUv2 (3-task) using SegNet (Badrinarayanan et al., 2017) as in Liu et al. (2019), and on Cityscapes (3-task) using DeepLabV3 (Chen, 2017) following Liu et al. (2022a). We also evaluate on CelebA, which contains 40 binary classification tasks, and adopt the CNN architecture used in Liu et al. (2024); Ban & Ji (2024). For pretrained CNNs, we integrate TS$\sigma$BN into LibMTL (Lin & Zhang, 2023) using DeepLabV3 with a pretrained ResNet50 backbone on NYUv2 (3-task) and Cityscapes (2-task). This allows comparison to a wide range of recent MTL baselines under a consistent framework.

**Vision Transformers.** We evaluate TS$\sigma$BN on two transformer-based MTL setups that reflect current state-of-the-art: MoE-style modulation, and parameter-efficient adapter-based methods. Both settings use pretrained Vision Transformer backbones with CNN based fusion or downsampling modules before task-specific decoders. For recent MoE MTL methods we follow the MLoRE protocol (Yang

et al., 2024) on PascalContext (5-task). We use a pretrained ViT-S backbone (Dosovitskiy et al., 2021) and fine-tune the entire model. We also evaluate TS$\sigma$BN on the MTLoRA benchmark (Agiza et al., 2024), which focuses on parameter-efficient MTL. This setup uses a partially frozen Swin-T (Liu et al., 2021c) backbone on PascalContext (4-task). We compare against a wide range of LoRA and adapter based models reported in MTLoRA. To showcase compatibility we also evaluate TS$\sigma$BN with added task-generic (shared) LoRA($r = 16$) adapters.

**Multi-task evaluation**. Following Maninis et al. (2019) to evaluate a multi-task model, we compute the average per-task performance gain or drop relative to a baseline $B$ specified in the top row of the results tables. $\Delta m\% = \frac{1}{T} \sum_{t=1}^{T} (-1)^{\delta_t} \frac{M_{m,t} - M_{B,t}}{M_{B,t}} \times 100$, where $M_{m,t}$ is the performance of a model $m$ on a task $t$, and $\delta_t$ is an indicator variable that is 1 if a lower value shows better performance for the metric of task $t$. All results are presented as an average over three independent runs. Additionally, we report parameters (P) and FLOPs (F) relative to the baseline.

**Baselines.** Across all experiments we compare TS$\sigma$BN to a set of standard and protocol-specific multi-task baselines. The most common reference points are Single-Task Learning (STL), which trains a separate model for each task, and Hard Parameter Sharing (HPS), which shares the entire backbone with equal task weights. We also include TSBN, the multi-task equivalent of domain-specific BN, which simply duplicates BN layers without our reparameterization and optimization changes. Each experimental setting includes additional baselines that follow the protocol and architecture family, reflecting standard practice in prior work and ensuring fair comparisons. For completeness, we also report results for multi-task optimization methods in the Appendix G.

| Method | NYUv2 | | | | | Cityscapes | | | | | CelebA | | |
|---|---|---|---|---|---|---|---|---|---|---|---|---|---|
| | #P | Seg↑ | Depth↓ | Norm↓ | Δ% | #P | Seg↑ | P.Seg↑ | Disp↓ | Δ% | #P | F1↑ | Δ% |
| STL | 1.00 | 41.45 | 0.580 | 23.80 | 0.00 | 1.00 | 56.61 | 53.95 | 0.841 | 0.00 | 1.00 | 68.21 | 0.00 |
| HPS | 0.33 | 42.17 | 0.502 | 26.63 | +1.07 | 0.60 | 55.03 | 51.92 | 0.796 | -0.39 | 0.03 | 67.06 | -1.69 |
| CS | 1.00 | 41.77 | 0.492 | 26.15 | +1.98 | 1.00 | 56.73 | 53.89 | 0.781 | +2.43 | 1.01 | 65.57 | -3.86 |
| MTAN | 0.59 | 43.12 | 0.508 | 25.44 | +3.14 | 0.78 | 55.83 | 52.61 | 0.799 | +0.39 | 0.39 | 59.49 | -12.78 |
| TSBN | 0.33 | 43.47 | 0.494 | 25.32 | +4.42 | 0.61 | 56.10 | 52.82 | 0.806 | +0.40 | 0.03 | 67.17 | -1.52 |
| **TS$\sigma$BN** | **0.33** | 43.75 | 0.484 | 24.09 | **+6.93** | **0.60** | 56.45 | 53.26 | 0.814 | +0.57 | **0.03** | 69.45 | **+1.81** |

Table 1: Comparison of encoder-based soft-sharing architectures on NYUv2 (3-task SegNet), Cityscapes (3-task DeepLabV3), and CelebA (40-task CNN) trained from random initialization. TS$\sigma$BN achieves the best overall performance on NYUv2 and CelebA by a significant margin, and competitive results on Cityscapes, while maintaining the lowest parameter count.

| Method | NYUv2 | | | | | | CityScapes | | | | |
|---|---|---|---|---|---|---|---|---|---|---|---|
| | #P | #F | Seg↑ | Depth↓ | Normal↓ | Δ% | #P | #F | Seg↑ | Depth↓ | Δ% |
| HPS | 1.00 | 1.00 | 53.93 | 0.3825 | 23.57 | 0.00 | 1.00 | 1.00 | 69.81 | 0.0125 | 0.00 |
| CS | 1.65 | 1.69 | 53.44 | 0.3818 | 23.15 | +0.35 | 1.42 | 1.44 | 69.97 | 0.0123 | +0.55 |
| MMOE | 1.35 | 1.34 | 53.14 | 0.3876 | 23.02 | -0.15 | 1.42 | 1.44 | 69.81 | 0.0126 | -0.43 |
| MTAN | 1.28 | 1.56 | 54.64 | 0.3771 | 23.12 | +1.55 | 1.29 | 1.48 | 70.62 | 0.0125 | +0.49 |
| CGC | 2.01 | 2.03 | 53.27 | 0.3914 | 22.14 | +0.84 | 1.85 | 1.88 | 69.75 | 0.0125 | -0.12 |
| PLE | 2.41 | 2.71 | 52.75 | 0.3943 | 22.10 | +0.32 | 1.95 | 2.32 | 69.30 | 0.0129 | -2.02 |
| LTB | 1.65 | 1.69 | 52.58 | 0.3828 | 23.31 | -0.49 | 1.42 | 1.44 | 69.81 | 0.0125 | -0.35 |
| DSelect-k | 1.38 | 1.34 | 53.75 | 0.3802 | 23.18 | +0.64 | 1.44 | 1.44 | 69.67 | 0.0124 | +0.26 |
| TSBN | 1.00 | 1.69 | 53.44 | 0.3761 | 23.01 | +1.04 | 1.00 | 1.44 | 69.89 | 0.0124 | +0.38 |
| **TS$\sigma$BN** | **1.00** | 1.69 | 53.78 | 0.3735 | 22.31 | **+2.48** | **1.00** | 1.44 | 70.17 | 0.0123 | **+0.85** |

Table 2: Comparison of various multi-task architectures within the LibMTL framework using DeepLabV3 with a pre-trained ResNet-50 backbone on NYUv2 (3-task) and CityScapes (2-task). TS$\sigma$BN achieves the best overall performance while being the most parameter-efficient.

## 6.1 RESULTS

Across all experimental settings, TS$\sigma$BN delivers consistent gains in performance while maintaining superior parameter efficiency.

On randomly initialized CNNs in Table 1, TS$\sigma$BN achieves the best results on NYUv2 (+6.93%) and CelebA (+1.81%), with competitive performance on Cityscapes, all at the lowest parameter cost. Notably, soft parameter sharing methods underperform the STL baseline on CelebA, highlighting their poor scalability to

| Method | Seg. mIoU↑ | Parts. mIoU↑ | Sal. maxF↑ | Norm. mErr↓ | Bdry. odsF↑ | #F (G) | #P (M) |
|---|---|---|---|---|---|---|---|
| M$^3$ViT | 72.80 | 62.10 | 66.30 | 14.50 | 71.70 | 420 | 42 |
| Mod-Squad | 74.10 | 62.70 | 66.90 | 13.70 | 72.00 | 420 | 52 |
| TaskExpert | 75.04 | 62.68 | 84.68 | 14.22 | 68.80 | 204 | 55 |
| MLoRE | 75.64 | 62.65 | 84.70 | 14.43 | 69.81 | 72 | 44 |
| TSBN | 75.95 | 63.33 | 84.655 | 14.16 | 68.05 | 214 | 29 |
| **TS$\sigma$BN** | **77.12** | **64.73** | **85.24** | 14.04 | 70.00 | 214 | **29** |

Table 3: PascalContext results for MoE-style models using a pretrained ViT-S backbone. TS$\sigma$BN delivers best results using fewer parameters.

many tasks, whereas TS$\sigma$BN remains robust. On pretrained CNNs within LibMTL in Table 2, TS$\sigma$BN achieves the strongest overall performance on both NYUv2 (+2.48%) and Cityscapes (+0.85%), outperforming all MTL baselines, including MoE approaches, while remaining lightweight. On pre-trained transformers with ViT-S in Table 3, TS$\sigma$BN surpasses state-of-the-art methods, such as M$^3$ViT, Mod-Squad, and MLoRE, while using fewer parameters. Relative to other parameter-efficient fine-tuning approaches in Table 4 TS$\sigma$BN offers the best performance relative to its trainable parameter count. Adding shared capacity via LoRA($r = 16$) adapters further improves performance.

We note that even the simpler TSBN variant (without sigmoid and differential learning rates) delivers competitive performance out of the box, suggesting that complex architectures may be unnecessarily over-engineered. Overall, TS$\sigma$BN achieves the best balance of accuracy, efficiency, and simplicity, consistently outperforming specialized MTL architectures across CNNs and transformers, while scaling to many-task regimes.

## 7 ABLATIONS

### 7.1 DISCRIMINATIVE LEARNING RATES

We analyze the impact of different learning rate multipliers applied to the $\sigma$BN layers, focusing on their effect on the distribution of scaling parameters $\gamma_t$ and overall model performance. Figure 5 illustrates how varying the $\alpha_{BN}$ multiplier influences the distribution of $\sigma(\gamma_t)$ values across all filters. A more detailed task-wise breakdown is provided in the Appendix. Higher learning rates induce more significant parameter variance, increasing their expressivity. Since $\sigma(\gamma_t)$ is initialized at $0.5$, lower learning rates result in minimal divergence, with $\alpha_{\sigma BN} = 1$ being excluded as it shows almost no differentiation between tasks. At $\alpha_{\sigma BN} = 100$, we see a substantial spread in $\sigma(\gamma_t)$ values across the full $[0, 1]$ range, allowing tasks to choose and specialize on subsets of filters. However, an extreme learning rate of $\alpha_{\sigma BN} = 10^3$ leads to a highly polarized distribution, where filter importances collapse to a binary mask, effectively enforcing a hard-partitioning regime. These findings highlight how BN learning rates control the

| Method | Seg. mIoU↑ | Parts mIoU↑ | Sal. mIoU↑ | Norm. mErr↓ | Δm (%) | #P (M) |
|---|---|---|---|---|---|---|
| STL | 67.21 | 61.93 | 62.35 | 17.97 | 0 | 112.62 |
| HyperFormer | 71.43 | 60.73 | 65.54 | 17.77 | 2.64 | 72.77 |
| MTL-Full FT | 67.56 | 60.24 | 65.21 | 16.64 | 2.23 | 30.06 |
| Adapter | 69.21 | 57.38 | 61.28 | 18.83 | -2.71 | 11.24 |
| Polyhistor | 70.87 | 59.15 | 65.54 | 17.77 | 2.34 | 8.96 |
| MTLoRA(r=16) | 68.19 | 58.99 | 64.48 | 17.03 | 1.35 | 4.95 |
| VL-Adapter | 70.21 | 59.15 | 62.29 | 19.26 | -1.83 | 4.74 |
| **TS$\sigma$BN(r=16)** | **70.00** | **58.01** | **63.89** | **16.85** | **1.63** | **4.25** |
| VPT-deep | 64.35 | 52.54 | 58.15 | 21.07 | -10.85 | 3.43 |
| MTLoRA+(r=8) | 68.54 | 58.30 | 63.57 | 17.41 | 0.29 | 3.15 |
| **TS$\sigma$BN** | **69.38** | **57.46** | **63.74** | **17.00** | **0.91** | **3.08** |
| LoRA | 70.12 | 57.73 | 61.90 | 18.96 | -2.17 | 2.87 |
| BitFit | 68.57 | 55.99 | 60.64 | 19.42 | -4.60 | 2.85 |
| Compacter | 68.08 | 56.41 | 60.08 | 19.22 | -4.55 | 2.78 |
| Compacter++ | 67.26 | 55.69 | 59.47 | 19.54 | -5.84 | 2.66 |
| VPT-shallow | 62.96 | 52.27 | 58.31 | 20.90 | -11.18 | 2.57 |

Table 4: Results for PEFT baselines using a Swin-T backbone on PascalContext sorted by number of trainable parameters. TS$\sigma$BN and its combination with LoRA($r = 16$) deliver the best performance relative to their size.

degree of task-specific capacity allocation, influencing both representation disentanglement and network adaptability.

We further analyze the impact of different learning rate multipliers on the MTL performance in Table 6. For TSBN, moderate multipliers yield small gains, but performance collapses at high rates. In contrast, $\sigma$BN consistently benefits from larger multipliers across values, indicating that sigmoid activation is essential both for unlocking greater improvements and for robustness.

## 7.2 ROBUSTNESS TO LOSS SCALES

A well-known challenge in multi-task learning is the discrepancy in loss scales and, consequently, gradient magnitudes across tasks, which can lead to task dominance and suboptimal performance. Many existing approaches rely on manual tuning or specialized optimization strategies for dynamic weighting. Our method is highly robust to perturbations of loss scales without any additional changes.

To evaluate the robustness of our method to loss weight perturbations, we conduct a series of experiments on NYUv2 by varying the weight of each task. Specifically, we scale each task loss by factors of $\{0.5, 1.5, 2.0\}$ while maintaining the default weight of 1.0 for the remaining tasks. The distribution of relative performances under these perturbations is visualized in Figure 5. TS$\sigma$BN shows the lowest variance under loss scale perturbations, indicating robustness to task dominance and improved optimization stability.

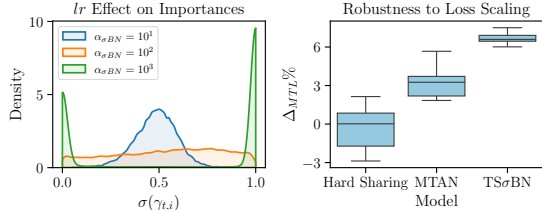

Figure 5: Effect of BN-specific learning rate multipliers on the $\sigma(\gamma_t)$ filter importances distribution (left) and relative performance of models under loss scale perturbations (right).

| $\alpha_{\sigma BN}$ | $10^0$ | $10^1$ | $10^2$ | $10^3$ |
|---|---|---|---|---|
| TSBN | +4.09% | +4.80% | +4.42% | -2.96% |
| TS$\sigma$BN | +4.02% | +5.67% | +6.93% | +4.33% |

Figure 6: Impact of different BN specific learning rate multipliers on the performance of TSBN and TS$\sigma$BN relative to STL on NYUv2.

## 8 CONCLUSION

We present TS$\sigma$BN, a simple soft-sharing mechanism for multi-task learning that relies only on task-specific normalization layers. Using a sigmoid-gated reparameterization and differential learning rates, our method turns BN from a normalization module into a stable and expressive tool for capacity allocation and interference reduction.

Across convolutional and transformer architectures, TS$\sigma$BN achieves competitive or superior performance while using substantially fewer parameters. Notably, it matches or outperforms state-of-the-art MoE-style and PEFT-based MTL methods without adding routing modules, experts, or adapters. The learned gates also provide a direct view of model behavior, yielding interpretable measures of capacity allocation, filter specialization, and task relationships.

Overall, our results show that lightweight, normalization-driven designs can replace much heavier mechanisms while offering clearer interpretability. We hope this encourages a reevaluation of complexity in MTL and promotes simple, transparent alternatives.

ETHICS STATEMENT

This work does not involve human subjects, private data, or sensitive content. All datasets used (NYUv2, Cityscapes, CelebA, PascalContext) are publicly available and widely adopted benchmarks.

REPRODUCIBILITY STATEMENT

We provide comprehensive experimental details in the main text in Section 6 and Appendix F, including datasets, architectures, training protocols, and evaluation metrics.

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
