## A    INTERFERENCE

To further investigate task interference, we expand on the analysis presented in Section 4 and provide a more comprehensive view of gradient conflicts across all task pairs for NYUv2. Specifically, in figure 7 we plot the distribution of cosine similarities between gradients for every task pair across the shared parameters of the SegNet backbone.

In addition to the methods discussed in the main paper, we include Task-Specific Batch Normalization (TSBN) as a baseline. Interestingly, TSBN alone is sufficient to induce a mode around orthogonality, demonstrating that normalization can already reduce some degree of task interference. However, incorporating $\sigma$BN significantly amplifies this effect, further increasing the number of near-orthogonal gradients and reducing interference. This highlights the role of $\sigma$BN in not only mitigating conflicts but also improving gradient disentanglement across tasks.

It is important to note that the presented gradient distributions are measured after one epoch of training over the training set. As training progresses, we observe that the differences between methods become less pronounced. Regardless of the initial distribution, all approaches gradually converge toward a bell-shaped distribution centered around orthogonality. This suggests that while early-stage interference may impact optimization dynamics, multi-task models eventually adjust to reduce conflicts over time.

A notable exception is observed in MTAN, which produces more aligned gradients specifically for the semantic segmentation and surface normal estimation task pair. Despite this alignment, we do not observe a corresponding performance gain. This suggests that while reducing conflicts is beneficial, not all aligned gradients lead to improved task synergy, underscoring the notion that mitigating interference alone does not guarantee optimal performance.

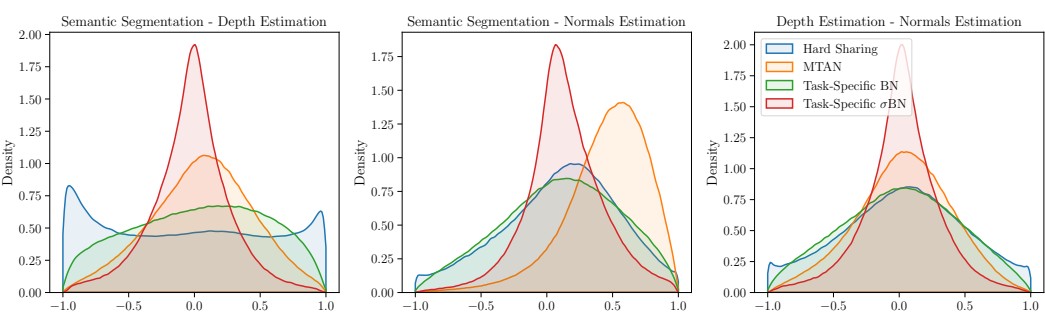

Figure 7: Distribution of gradient cosine similarities between all task pairs on the NYUv2 dataset using a SegNet backbone.

## B    DISENTANGLED TASK REPRESENTATIONS

We extend Figure 2 from Section 4 by visualizing encoder representations for all 40 tasks in the CelebA setting. As before, we use t-SNE to project the high-dimensional representations into a more interpretable space. Each data point is assigned representations for every task due to the nature of the soft parameter sharing paradigm, resulting in multiple embeddings per sample. In Figure 8, we observe that most tasks form well-separated clusters, though a few outliers exhibit some degree of overlap.

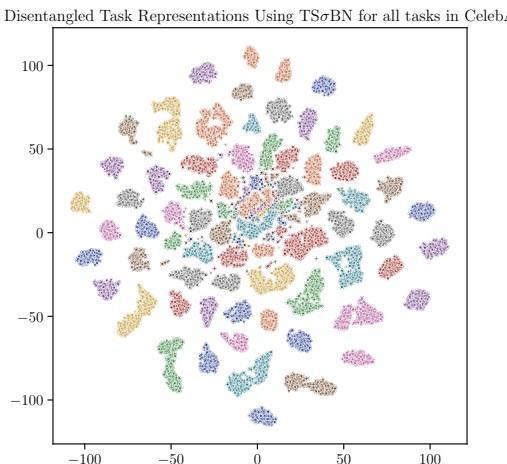

Figure 8: t-SNE visualization for all task representations for 1000 inputs from the CelebA dataset.

## C  ROBUST TASK RELATIONSHIPS

We utilize the CelebA dataset to identify relationships and hierarchies among the 40 binary classification tasks of facial attributes. We compute pairwise cosine similarities between task importance vectors, producing a task similarity matrix $S = \left[ \frac{I_i \cdot I_j}{\|I_i\| \, \|I_j\|} \right]_{i,j \in \mathcal{T}}$, that serves as the foundation for identifying task clusters and hierarchies. Crucially, for these relationships to be useful, they must be robust - unstable hierarchies would offer little insight into model behavior or optimization dynamics. We find the relationships from TS$\sigma$BN to be highly stable, with an average Spearman rank correlation of 0.8 between similarity matrices from seven independent training runs.

For a qualitative assessment of task relationships, we compute representative clusters of tasks from the seven runs. To achieve this, we construct a co-occurrence matrix that captures the frequency with which each pair of tasks appears in the same cluster. This co-occurrence matrix effectively aggregates clustering information from all runs, highlighting task pairs that consistently exhibit strong relationships regardless of initialization. We then apply hierarchical clustering directly to this matrix to identify a representative cluster of tasks that frequently co-occur.

The identified clusters exhibit apparent semantic coherence, as shown in Table 5. Since these clusters are derived from filter-usage based relationships, tasks grouped tend to rely on similar specialized filters within the network. This suggests that the model internally organizes tasks based on shared feature representations. Notably, the clustering patterns appear to correlate with the spatial proximity of facial attributes. For instance, tasks related to hair characteristics (e.g., Bangs, Blond Hair) form a distinct cluster. In contrast, facial hair attributes (e.g. Goatee, Mustache) are grouped separately, indicating that the network leverages localized feature detectors. This spatial coherence reinforces the idea that task relationships emerge from shared activations of filters sensitive to specific facial regions, reflecting the model's ability to capture both semantic and structural commonalities across tasks.

## D  DISCRIMINATIVE LEARNING RATES

We extend the ablation study from Section 7.1, investigating the impact of discriminative learning rates for $\sigma$BN layers. Specifically, we apply a higher learning rate to BN parameters, allowing them to adapt more rapidly to the shared convolutional layers before those layers undergo significant updates. This adjustment is controlled by a multiplier applied to the model's base learning rate.

In this more detailed analysis, we examine the importance distributions of filters per task across different learning rate multipliers. Figure 9 presents the resulting distributions for four multiplier values: $10^0, 10^1, 10^2, 10^3$. As the multiplier increases, the variance of filter importance distributions grows, leading to progressively softer filter allocations. At a multiplier of 1, BN parameters remain close to their initialization, resulting in near-uniform filter sharing across tasks, similar to hard

| # | Attributes |
|---|---|
| 1 | High Cheekbones, Mouth Slightly Open, Smiling |
| 2 | Bangs, Black Hair, Blond Hair, Brown Hair, Gray Hair, Straight Hair, Wavy Hair, Wearing Hat |
| 3 | Attractive, Bags Under Eyes, Big Nose, Young |
| 4 | Bald, Chubby, Double Chin, Receding Hairline, Wearing Necktie |
| 5 | Blurry, Heavy Makeup, Male, Pale Skin, Wearing Lipstick |
| 6 | 5 o'Clock Shadow, Goatee, Mustache, No Beard, Sideburns |
| 7 | Arched Eyebrows, Bushy Eyebrows, Narrow Eyes |
| 8 | Eyeglasses, Rosy Cheeks |
| 9 | Big Lips, Oval Face, Pointy Nose |
| 10 | Wearing Earrings, Wearing Necklace |

Table 5: Clusters of attributes extracted from a TS$\sigma$BN model trained on the 40-task CelebA dataset. Task relationships correlate with the spatial proximity of facial features, suggesting that the model organizes tasks based on localized filter activations, capturing both semantic and structural similarities.

parameter sharing. On the opposite extreme, a multiplier of $10^3$ effectively induces a binary filter mask, resembling a hard partitioning approach. Notably, $\sigma$BN plays a crucial role in stabilizing this process, as its sigmoid activation mitigates potential gradient explosion. We use $\alpha_{\sigma BN} = 10^2$ in all our experiments.

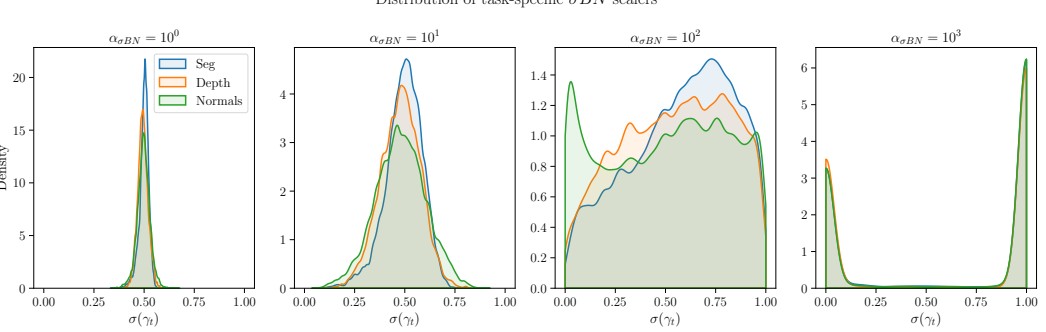

Figure 9: Detailed visualization of the effect of different learning rates on the distribution of task-specific $\sigma BN$ scaling parameters.

# E  EFFECTS OF TASK DIFFICULTY ON CAPACITY ALLOCATION

To further investigate MTL capacity allocation using the TS$\sigma$BN framework, we conduct a synthetic experiment designed to control task difficulty and relationships systematically. Specifically, we modify the NYUv2 dataset by removing the surface normals estimation task and replacing it with a noisy variant of the depth estimation task. We generate a family of datasets where the additional depth task is corrupted by Gaussian noise of increasing variance. Formally, given the original depth labels $D$, we construct synthetic tasks:

$$\tilde{D}_\xi = D + \mathcal{N}(0, \xi * \sigma_D^2), \tag{6}$$

where $\xi$ controls the level of corruption as a scaler of the original depth task's variance. Using TS$\sigma$BN, we analyze how model capacity is allocated between shared and task-specific components, as well as how task relationships change, by computing cosine similarity over task importance vectors.

In figure 10 we plot the decomposed task capacities and pairwise similarities for datasets with $\xi$ ranging between $[0, 3]$. As expected, when $\xi$ is low, the original and noisy depth tasks exhibit strong

alignment, reinforcing high shared capacity. However, as $\xi$ increases, the similarity between the tasks decreases, and their filter allocations become more distinct, with independent capacity increasing. This aligns with our hypothesis that related tasks co-adapt to share resources, whereas unrelated tasks require greater specialization. Overall, this experiment highlights how TS$\sigma$BN automatically balances shared and independent capacity in response to increasing task difficulty and lower task similarity.

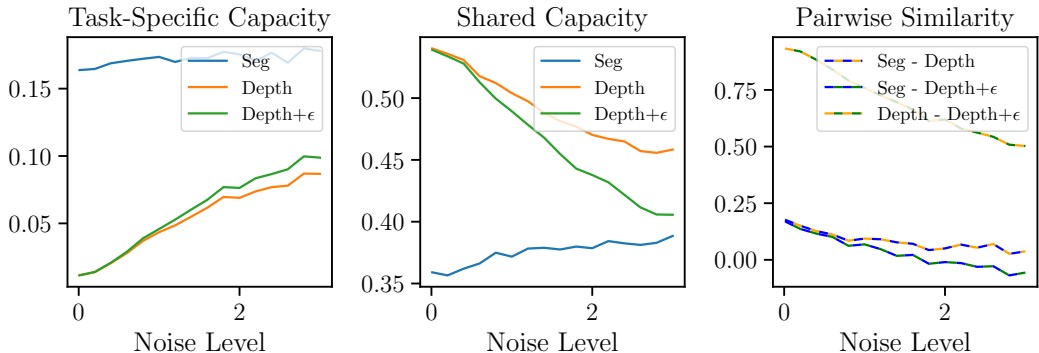

Figure 10: The effect of increasing task difficulty and decreasing similarity on capacity allocation in TS$\sigma$BN. For the noise scaling factor, a synthetic depth estimation task is generated with additive Gaussian noise. (Left) Independent task-specific capacities. (Middle) Shared capacity between tasks. (Right) Pairwise cosine similarity between task importance vectors.

## F    EXPERIMENTAL SETTINGS

**Hardware.** Experiments on NYUv2 and Cityscapes were run on an NVIDIA RTX 3090 GPU. Due to higher memory requirements, CelebA (40 tasks) and transformer-based models were trained on an NVIDIA A100 GPU.

### F.1    CNNs WITH RANDOM INITIALIZATION

**NYUv2.** We follow the setup of Liu et al. (2019; 2024) for base architecture, training configuration, and evaluation metrics. A multi-task SegNet is used, with both encoder and decoder shared across tasks and lightweight task-specific heads composed of two convolutional layers. All methods are trained with Adam ($lr = 10^{-4}$), using a step schedule that halves the learning rate at epoch 100. Training runs for 200 epochs with a batch size of 4.

**Cityscapes.** Following Liu et al. (2022a), we use DeepLabV3 with a ResNet-50 backbone and task-specific ASPP decoders, which account for most of the parameters. Optimization is performed with SGD ($lr = 10^{-2}$, weight decay $= 10^{-4}$, momentum $= 0.9$) for 200 epochs using a CosineAnnealing scheduler and batch size of 4. For TS$\sigma$BN layers, weight decay is disabled.

**CelebA.** We adopt the configuration from Liu et al. (2024); Ban & Ji (2024), using a shared CNN backbone with task-specific linear classifiers. Models are trained for 15 epochs with Adam ($lr = 3 \times 10^{-4}$) and batch size 256.

### F.2    CNNs WITH PRETRAINED WEIGHTS

**Implementation.** Converting pretrained BN layers into $\sigma$BN depends on their weights. A network trained from scratch may learn a purely linear transformation, but converting an affine layer to linear is not possible unless $\beta = 0$. To avoid conversion shock, we copy the pretrained biases but keep them frozen during training. In ResNet-50 pretrained on ImageNet, most BN scale parameters ($\gamma$) fall within (0,1), allowing them to be represented by the sigmoid function. We therefore apply the inverse sigmoid to initialize $\sigma$BN scales, ensuring consistency with the pretrained distribution.

**NYUv2.** We follow the default LibMTL configuration (Lin & Zhang, 2023), reporting results of related methods as published. Models are trained with Adam ($lr = 10^{-4}$) for 200 epochs, using StepLR with $\gamma = 0.5$ at epoch 100 and a batch size of 4.

**Cityscapes.** Same as above, except the batch size is set to 16 due to memory constraints. All results, including related methods, are averaged over three random seeds for fair comparison.

### F.3 VISION TRANSFORMERS

Following prior work, we extract intermediate representations from the transformer backbone and process them through a lightweight multi-scale fusion module. The module consists of four Conv–TS$\sigma$BN–GELU blocks shared across tasks, implemented as two 1×1 convolutions for channel adjustment squeezing two 3×3 convolutions with width 512; decoder inputs have width 196. In the ViT patch embedding, we replace the normalization with a $\sigma$LN layer. All remaining LNs in the backbone are converted to TSLN, since their pretrained scales often exceed the sigmoid co-domain.

Following the MLoRE setup (Yang et al., 2024), we train a ViT-S backbone using Adam with base learning rate $2 \times 10^{-5}$ and polynomial decay. Learning-rate multipliers of 100 and 10 are applied to TS$\sigma$BN and TSLN/TS$\sigma$LN layers respectively. Dropout and DropPath are disabled. Models are trained for 60k iterations.

### F.4 SWIN TRANSFORMER

We follow the MTLoRA (Agiza et al., 2024) experimental protocol and use a pretrained Swin-T backbone. Intermediate representations from each stage are extracted and passed through a downsampling module before the task-specific HRNet heads. The original downsampler consists of a single 1×1 convolution per scale; in our variant, we insert TS$\sigma$BN modules both before and after this convolution. As in the ViT experiments, all remaining LayerNorm layers in the backbone are converted to TSLN.

In the PEFT setting most of the Swin-T backbone remains frozen, including the self-attention and MLP blocks. The patch embedding and patch merging layers stay trainable. When combining TS$\sigma$BN with LoRA, we add adapters only to the frozen linear layers and keep them shared across tasks, so they do not contribute task-specific capacity. We integrate our model using the MTLoRA codebase and keep all training hyperparameters unchanged. The only modification is the optimizer configuration required to support differential learning rates: as in all experiments we use a multiplier of 100 for TS$\sigma$BN parameters and 10 for LN.

## G ADDITIONAL BENCHMARKS

For completeness, we benchmark TS$\sigma$BN against multi-task optimization (MTO) methods. These approaches are orthogonal to architectural soft-sharing: instead of allocating task-specific parameters, they operate on a fully shared backbone and modify the optimization dynamics to mitigate conflict.

MTO methods fall into two major categories: loss-balancing strategies (e.g., UW, DWA, RLW) and gradient manipulation strategies (e.g., GradNorm, MGDA, PCGrad, CAGrad, Nash-MTL). Because they maintain a fully shared backbone, these methods are often highly parameter-efficient, but this comes with different trade-offs: they typically require additional computational overhead, involve external solvers to compute descent directions, and can significantly increase training time.

We evaluate TS$\sigma$BN within the LibMTL framework, using the NYUv2 setup with a pretrained DeepLabV3–ResNet50 backbone. We compare against the suite of optimization-based methods available in LibMTL, including UW (Kendall et al., 2017), GradNorm (Chen et al., 2018), MGDA (Sener & Koltun, 2018), DWA (Liu et al., 2019), GLS (Chennupati et al., 2019), PCGrad (Yu et al., 2020), GradDrop (Chen et al., 2020), IMTL (Liu et al., 2021b), GradVac (Wang et al., 2021), CAGrad (Liu et al., 2021a), Nash-MTL (Navon et al., 2022), and RLW (Lin et al., 2021).

Table 6 reports the results. TS$\sigma$BN achieves the highest overall multi-task gain among all optimization-based methods on top of the same HPS architecture, while maintaining the simplicity of a purely architectural soft-sharing mechanism.

| Method | Seg↑ | Depth↓ | Normal↓ | Δm (%)↑ |
|---|---|---|---|---|
| HPS | 53.93 | 0.3825 | 23.57 | 0.00 |
| +GradNorm | 53.91 | 0.3842 | 23.17 | 0.41 |
| +UW | 54.29 | 0.3815 | 23.48 | 0.44 |
| +MGDA | 53.52 | 0.3852 | 22.74 | 0.69 |
| +DWA | 54.06 | 0.3820 | 23.70 | -0.06 |
| +GLS | 54.59 | 0.3785 | 22.71 | 1.97 |
| +PCGrad | 53.94 | 0.3804 | 23.52 | 0.26 |
| +GradDrop | 53.73 | 0.3837 | 23.54 | -0.19 |
| +IMTL | 53.63 | 0.3868 | 22.58 | 0.84 |
| +GradVac | 54.21 | 0.3859 | 23.58 | -0.14 |
| +CAGrad | 53.97 | 0.3885 | 22.47 | 1.06 |
| +Nash-MTL | 53.41 | 0.3867 | 22.57 | 0.73 |
| +RLW | 54.04 | 0.3827 | 23.07 | 0.76 |
| **TS$\sigma$BN** | 53.78 | **0.3735** | **22.30** | **2.48** |

Table 6: Comparison with optimization-based MTL methods on the LibMTL NYUv2 benchmark using a pretrained DeepLabV3–ResNet50 backbone. All optimization methods are applied on top of the same HPS architecture. TS$\sigma$BN achieves the highest overall multi-task improvement among all optimization-based approaches.