# OpenReview forum: "Simplifying Multi-Task Architectures Through Task-Specific Normalization"
_ICLR.cc/2026/Conference — Submitted to ICLR 2026_

### Official Review · Reviewer_BXZU · 2025-10-21

**Soundness:** 3
**Presentation:** 3
**Contribution:** 2
**Rating:** 6
**Confidence:** 4

**Summary:**

The paper proposes a  task‑Specific Sigmoid Batch Normalization (TSσBN) approach for multi-task learning, which introduces per‑task scaling parameters that are passed through a sigmoid gating mechanism to softly allocate network capacity. The authors method isolates BatchNorm as the sole soft-sharing mechanism, showing that it is a sufficient solution for competitive MTL while challenging unnecessary complexity of previous MTL approaches. Experiments confirm the applicability of the method and its low computational requirements.

**Strengths:**

- The capacity TS$\sigma$BN to built an importance matrix is a highly appreciated feature of the method. It allows interpretable insights into the model behaviour.
- TS$\sigma$BN robustness to loss scales without additional changes is another strength.
- The idea is quite simple but is an important contribution to MTL. The concept of domain-specific batch layer norm while not new for MTL show surprising positive effects. Besides that the TS$\sigma$BN is easy to implement and require minimal changes to current architectures.
- The wide range of experimental architectures tested show that the method can be used on multiple vision domains.

**Weaknesses:**

- While the authors focused quite extensively on vision tasks, it is unclear whether TS$\sigma$BN generalizes beyond computer vision. Small experiments on NLP or time series MTL (MIMIC-III) tasks would strength the paper claim on generalisation of the method.
- The paper provides mainly empirical evidences. A deeper theoretical justification of why task-specific batch normalization disentagle representations could make the work more compelling.
- Another aspect of MTL methods is its hyper-parameter tuning sensitivity. While the authors provide ablations, these remain a risk that performance is sensitive to this $\sigma$ parameter or dataset-specific tuning.

**Questions:**

- Regarding broader domain evaluation, the authors can provide experiments or at least discuss how TS$\sigma$BN will perform in other domains.
- Can the authors provide guidance on selecting the $\sigma$ hyper-parameter and the threshold $\tau$ for filter specialization.
- I suggest the author to include some optimization-based methods to the comparison (GradNorm and CAGrad) to demonstrate how TS$\sigma$BN compares relatively to techniques that modify gradients rather than architectures.

---

> ### Author Response · Authors · 2025-11-17
>
> We thank the reviewer for the constructive and thoughtful feedback. Below we address each point.
>
> ### **1. Additional Experiments**
>
> Our evaluation follows the norms established by recent multi-task learning papers such as M3ViT, MLoRE, MTLoRA, and Mod-Squad, all of which focus exclusively on computer vision, as the dominant track for architectural MTL research.
>
> While we agree that cross-domain studies are valuable, **our experimental coverage is already substantially broader than standard practice**. With the additional experiments included in the revision, we now evaluate on **four datasets** (NYUv2, Cityscapes 2- and 3-task, CelebA, PascalContext 4- and 5-task), **five backbone architectures** (SegNet, DeepLabV3-ResNet50, CNN, ViT-S, Swin-T) and **seven total settings** (pretrained and from-scratch). Recent MTL works typically evaluate on one or two datasets with one backbone.
>
> Evaluating NLP or time-series tasks would require domain-specific training pipelines and baselines, which are outside the scope of this vision-focused paper. Importantly, TSσBN is applicable to any architecture containing normalization layers, and we will emphasize this generality in the revision.
>
> ### **2. Additional Justification**
>
> We appreciate the reviewer’s interest in deeper theoretical insight. Our contribution is primarily empirical, but it is **grounded in established results** showing:
> (i) the expressive capacity of BN beyond normalization (Santurkar et al., 2018; Bjorck et al., 2018; Frankle et al., 2021), and
> (ii) the ability of BN to induce implicit orthogonalization and feature decoupling, which aligns with the task–filter specialization observed in our analysis (Daneshmand et al., 2021; Luo et al., 2019).
>
> Beyond this, a major contribution of our work is a novel **interpretability framework** allowing us to analyze MTL behaviour through capacity decompoistion, feature specialization and task relationships. These analyses jointly reveal why the method behaves well, acting as debugging tools that come "for free" when training a model. They provide **empirical evidence** that TSσBN produces results consistent with expected MTL behaviour. To our knowledge, no prior MTL work provides such a unified, analysis-oriented view across all three perspectives.
>
> ### **3. Hyper-parameter Sensitivity**
>
> A practical advantage of TSσBN is that it **requires no dataset-specific tuning**. The only hyperparameter is the learning-rate multiplier on γ. As shown in Table 6, performance is robust across a broad range $(10^0-10^3)$, and we fix it for σBN to 100 for all experiments without tuning.
> - σ is not a hyperparameter—it is the sigmoid applied to the learnable γ parameter.
> - τ is only used after training to identify filter specialization. It has no effect on training dynamics or model performance, and setting τ = 0.5 naturally encodes majority usage.
>
> Thus, TSσBN has substantially lower tuning burden than existing MTL methods.
>
>
> ### **4. MTO Baselines**
> Optimization-based MTL algorithms form a class of methods orthogonal to architectural approaches. They operate by modifying optimization dynamics (e.g., loss weighting, gradient manipulation) rather than producing task-specific representations. Because of this, they come with distinct trade-offs, notably strong parameter efficiency, but substantially longer training times and poor scalability to large models. Making a comprehensive comparison would require a deeper dedicated analysis taking into consideration these various aspects.
>
> Nonetheless, we agree that positioning TSσBN relative to this family of methods is valuable for broader context. In the LibMTL setting (pretrained DeepLabV3–ResNet50 on NYUv2), we compare our method against the published results for major optimization-based baselines. Under this standardized protocol, TSσBN achieves the highest overall multi-task performance among all these methods.
>
> We will include this comparison table in the appendix and note the potential composability of TSσBN with optimization-based techniques as an interesting direction for future work.
>
> | Method   | Seg ↑ | Depth ↓ | Normal ↓ | Δm ↑ (%) |
> |----------|-------|----------|-----------|-----------|
> | EW       | 53.93 | 0.3825   | 23.57     | 0.00     |
> | GradNorm | 53.91 | 0.3842   | 23.17     | 0.41     |
> | UW       | 54.29 | 0.3815   | 23.48     | 0.44     |
> | MGDA     | 53.52 | 0.3852   | 22.74     | 0.69     |
> | DWA      | 54.06 | 0.3820   | 23.70     | -0.06    |
> | GLS      | 54.59 | 0.3785   | 22.71     | 1.97     |
> | PCGrad   | 53.94 | 0.3804   | 23.52     | 0.26     |
> | GradDrop | 53.73 | 0.3837   | 23.54     | -0.19    |
> | IMTL     | 53.63 | 0.3868   | 22.58     | 0.84     |
> | GradVac  | 54.21 | 0.3859   | 23.58     | -0.14    |
> | CAGrad   | 53.97 | 0.3885   | 22.47     | 1.06     |
> | Nash-MTL | 53.41 | 0.3867   | 22.57     | 0.73     |
> | RLW      | 54.04 | 0.3827   | 23.07     | 0.76     |
> | **TSσBN**    | 53.78 | 0.3735   | 22.30     | **2.48** |

---

### Official Review · Reviewer_gf9H · 2025-10-31

**Soundness:** 2
**Presentation:** 2
**Contribution:** 1
**Rating:** 2
**Confidence:** 4

**Summary:**

In this paper, the authors propose an MTL architecture that shares all features for every task and allocates task-specific normalization layers. The "TSBN" variant replaces standard affine Batch Norm with a bounded sigmoid scalar per channel. An independent larger learning rates for BN parameters is used to learn capacity allocation faster. Evaluations are conducted on NYUv2, Cityscapes, and CelebA for CNN backbones, and Pascal-Context for ViT small. Experimental results show competitive accuracy vs. soft-sharing, MoE, and transformer MTL baselines at notably lower parameter cost. Finally, the per-task BN scales are used to analyze capacity sharing, task similarity, and specialization.

**Strengths:**

- The paper is clearly written and easy to understand.

- The paper addresses a persistent problem of negative interference in MTL. Finding efficient methods to mitigate this issue is of high importance to the field. The proposed method is simple and practical. It is also very straightforward for integration in practical settings.

- The ablations and analyses on task-filter importance analysis (Figure 4) are intuitive and match expectations.

**Weaknesses:**

This paper suffers from significant limitations in its current form, primarily concerning novelty, the depth of its empirical evaluation, and its positioning within the current literature. In its current form, this work extremely incremental and the core of the contributions around task-specific batch norms have already been established for 5< years.

**Limited Novelty:** The central contribution of this paper concerns using task-specific parameters or statistics in BatchNorm for MTL. Task-specific BN for MTL or multi-domain sharing is widely known and used for several years. Prior works and libraries explicitly advocate per-task BN, and domain-specific BN is standard for sharing all but BN across conditions, reducing the novelty of the core mechanism. For example Xie et al. [1,2] uses Task-Specific Batch Normalization for class incremental learning. The paper "Task Adaptive Parameter Sharing for Multi-Task Learning" [3] considers task-specific batch norm as a standard practice and considers it as a basic baseline. TaskNorm has been used for meta learning as well [4] in 2020. The paper fails to acknowledge this body of work sufficiently and does not clearly articulate what makes its specific implementation a novel advancement over these established techniques.

**Outdated Backbones**: The experiments are limited to very old architectures like ResNet-50 and ViT-S. To prove the method's relevance, it is crucial to demonstrate its effectiveness on current SoTA backbones such as ConvNeXt, DETR, etc. It is unclear if the performance gains would persist on these more powerful and architecturally different models.

**Benchmark scope:** The evaluation leans on legacy datasets such as NYUv2, Cityscapes, Pascal-Context, CelebA, which remain "widely-used" but are increasingly considered limited proxies in top venues. The authors should extend their experiments to include at least one large-scale, challenging benchmark such as Taskonomy.

**Outdated Related Works:** Reading the Intro and Related works section gives the impression the paper is behind the field for a few years. The paper fails to acknowledge the MTL research in research years. Most cited works are from 2017 to 2023.

[1] Xie, Xuchen, et al. "Class Incremental Learning with Task-Specific Batch Normalization and Out-of-Distribution Detection." arXiv preprint arXiv:2411.00430 (2024).

[2] Xie, Xuchen, et al. "Task-incremental medical image classification with task-specific batch normalization." Chinese Conference on Pattern Recognition and Computer Vision (PRCV). Singapore: Springer Nature Singapore, 2023.

[3] Wallingford, Matthew, et al. "Task adaptive parameter sharing for multi-task learning." Proceedings of the IEEE/CVF Conference on Computer Vision and Pattern Recognition. 2022.

[4] Bronskill, John, et al. "Tasknorm: Rethinking batch normalization for meta-learning." International Conference on Machine Learning. PMLR, 2020.

**Questions:**

- The Tables have abbreviations for method names that are never introduced. What is HPS?
- How does the parameter efficiency and accuracy of the method compare to a the modern methods which use LoRA adapters to specialize the network to each task?

---

> ### Author Response · Authors · 2025-11-17
> **Comparison to recent parameter-efficient baselines**
>
> We thank the reviewer for the thoughtful and constructive feedback. To directly address the concerns regarding benchmarking breadth and modern baselines, we have added a new experimental suite following the MTLoRA (2024) parameter-efficient benchmarks on Pascal-Context using a Swin-T backbone and HRNet heads. We integrate TSσBN into this framework both as a standalone modulation mechanism and in combination with shared (task-generic) LoRA adapters, and we compare against state-of-the-art parameter-efficient approaches.
>
> Below we provide a condensed version of the MTLoRA benchmark results, sorted by parameter count. TSσBN matches or surpasses the performance of larger baselines, demonstrating that our approach remains highly competitive even within modern PEFT. The full results table will be included in the revised paper.
>
>
> | Method                         | SemSeg↑ | HumanParts↑ | Saliency ↑ | Normals↓ | Δm↑ (%)   | Trainable #P (M) |
> | ------------------------------ | --------------- | -------------------- | ---------------- | ---------------- | -------- | -------------------- |
> | Single Task                | 67.21           | 61.93                | 62.35            | 17.97            | 0        | 112.62           |
> | MTL – Full Fine Tuning     | 67.56           | 60.24                | 65.21            | 16.64            | 2.23     | 30.06            |
> | Adapter                    | 69.21           | 57.38                | 61.28            | 18.83            | -2.71    | 11.24            |
> | Polyhistor                 | 70.87           | 59.15                | 65.54            | 17.77            | 2.34     | 8.96             |
> | MTLoRA (r=16)              | 68.19           | 58.99                | 64.48            | 17.03            | 1.35     | 4.95             |
> | VL-Adapter                 | 70.21           | 59.15                | 62.29            | 19.26            | -1.83    | 4.74             |
> | **TSσBN + LoRA(r=16)**     | 69.87           | 57.98                | 63.77            | 16.78            | **1.61** | 4.25             |
> | VPT-deep                   | 64.35           | 52.54                | 58.15            | 21.07            | -10.85   | 3.43             |
> | MTLoRA+ (r=8)              | 68.54           | 58.30                | 63.57            | 17.41            | 0.29     | 3.15             |
> | **TSσBN**                  | 69.38           | 57.46                | 63.74            | 17.00            | **0.91** | 3.08             |
> | LoRA                       | 70.12           | 57.73                | 61.90            | 18.96            | -2.17    | 2.87             |
> | BitFit                     | 68.57           | 55.99                | 60.64            | 19.42            | -4.60    | 2.85             |
> | Compactor                  | 68.08           | 56.41                | 60.08            | 19.22            | -4.55    | 2.78             |

---

> ### Author Response · Authors · 2025-11-17
> **Clarifications**
>
> We address the remaining points below. Some aspects of our contribution may not have come through due to space constraints; we appreciate the opportunity to clarify these misunderstandings and will incorporate the expanded explanations in the revision.
>
>
> ### **1. Novelty**
> We agree that BatchNorm has been reused in many contexts. Our contribution is not the idea of conditional BN itself, but *how* BN is used, *why* it works for single-domain MTL, and *what* this enables. We reposition BN from a normalization mechanism to a capacity-allocation and interference-mitigation mechanism in a setting where normalization is *not* the problem.
>
> - **Difference from the cited works.** As noted in L56–59 and L127–135, all works cited by the reviewer ([1–4]) address **domain-shift**. Continual Learning, Meta-Learning, Multi-Domain Learning are situations where the BN *normalization* step fails and statistics must be separated across domains or tasks. This motivation does *not* apply to single-domain MTL, where all tasks share the same input distribution and BN statistics are inherently compatible.
>
> - **Focus on transformation.** In our case, BN normalization does not need modification. Instead, we leverage the affine parameters (γ, β) to control task-specific feature usage, reduce interference, and allow implicit soft capacity allocation. This mechanism is absent from prior conditional BN work, which focuses solely on normalization robustness.
>
> - **TSσBN is not just task-specific BN.** The core mechanism is the **σBN reparameterization** as a interpretable gating function and optimization via **discriminative learning rates**. To our knowledge, no prior MTL or conditional BN paper uses this combination or demonstrates that BN layers alone can replace complex soft-sharing architectures.
>
> - **Novel interpretability.** TSσBN yields an *interpretable* task–filter importance matrix as a byproduct of training, enabling unified analysis of capacity sharing, task similarity, and filter specialization. Existing MTL work does not provide this level of interpretability or use BN parameters in this way.
>
> - **Novel positioning within modern MTL.** Recent MTL methods increasingly rely on heavy architectural mechanisms. Our contribution is the opposite: we show that **a minimal mechanism** can exceed the performance of SOTA architectures while being more parameter-efficient and easier to use.
>
>
> ### **2. Outdated Backbones**
> - **Broader coverage than other MTL papers.** Our experiments span **five architectures** across both pretrained and from-scratch settings. This is a **wider range** than most recent MTL papers, which typically evaluate on a single backbone family.
>
> - **Modern backbones are represented.** ViT-S and Swin-T are currently the dominant backbones in transformer-based MTL research (MLoRE 2024, MTLoRA 2024). We include all of these comparisons. To our knowledge there are currently no established MTL benchmarks on ConvNeXt or DETR, making them unsuitable for fair comparison.
>
> - **General applicability.** TSσBN applies to any architecture with normalization layers. Our results already demonstrate this generality across diverse backbone families.
>
>
> ### **3. Benchmark Scope**
> We respectfully disagree that benchmark scope is a weakness.
>
> - **Canonical datasets.** NYUv2, Cityscapes, Pascal-Context, and CelebA remain the standard MTL benchmarks in recent SOTA work because they provide aligned multi-label supervision and strong baselines. Taskonomy, while large, is rarely used for general MTL due to compute cost and lack of baselines.
>
> - **Broader coverage than recent papers.** Most recent MTL papers evaluate on one architecture and one or two datasets (MLoRE 2024, MTLoRA 2024 etc). In contrast, our work evaluates on **four datasets and five backbones**, covering **7 experimental setups**. This breadth is **larger than all cited works**, demonstrating robustness rather than narrow tuning.
>
> Thus, the evaluation scope is a strength, not a limitation.
>
>
> ### **4. Recency of References**
> - **Foundational works remain necessary.** MTL and BN have decades of accumulated literature. Foundational works from 2017–2020 define the concepts (gradient conflict, soft-sharing, BN expressivity and theory) that are essential to contextualize our contributions.
>
> - **Recent works are included.** Our related work section cites and discusses recent methods (2023+). Additionally, the new evaluation suite directly addresses the reviewer’s request to position TSσBN within the latest parameter-efficient MTL developments.
>
> We thank the reviewer for the detailed comments. We have addressed each concern, clarified the misunderstandings around novelty and positioning, and expanded our experiments with modern PEFT baselines, including results on Swin-T. With these additions, we believe the contribution, relevance, and empirical strength of the work are now clear, and we hope this resolves the reviewer’s concerns.

---

### Official Review · Reviewer_wFix · 2025-11-01

**Soundness:** 2
**Presentation:** 2
**Contribution:** 2
**Rating:** 4
**Confidence:** 2

**Summary:**

This paper proposes Task-Specific Sigmoid BatchNorm that swaps shared BN with per-task $\sigma$-BN to softly allocate capacity in shared backbones. Experiments (like Table 1 TS-$\sigma$-BN gets +6.93% over STL on NYUv2 with only 0.33× params) confirm the effectiveness of proporsed method on multiple benchmarks.

**Strengths:**

- The idea to make only normalization task-specific, without extra attention/routing, is simpler than works like Cross-Stitchwhich add task branches or dynamic sharing, so using $\sigma$-BN from (Suteu & Guo, 2022) for MTL is a nice, underexplored reuse.
- The reviewer found it interesting that TS-$\sigma$-BN stays stable even when they boost BN learning rates to 10² (Figure 6), while plain TSBN collapses, so the bounded gate actually matters.
- Writing is mostly clear.

**Weaknesses:**

- It should have been easy to show TSσBN vs MTAN vs MoE on CelebA with 40 tasks under the LibMTL setup (authors only report LibMTL for NYUv2/Cityscapes), and an ablation against simpler DSBN/TaskNorm baselines is missing.
- The reviewer questions the novelty of the contributions w.r.t. ICLR standards.
- Typos - Line 289: “respresentative” → “representative”

**Questions:**

The reviewer has one question:
- In Table 2 (LibMTL, NYUv2/Cityscapes) TSσBN uses 1.00× params and 1.69× FLOPs but TSBN also lists 1.69× FLOPs; can you clarify whether FLOPs include per-task heads or only BN copies, since this seems lower than MTAN’s added convolutions?

---

> ### Author Response · Authors · 2025-11-17
>
> We thank the reviewer for the constructive feedback. Below we clarify misunderstandings and address each concern.
>
> ### 1. **CelebA on LibMTL**
>
> - CelebA is not part of the LibMTL benchmark suite; we build on the setup from FAMO/FairGrad which uses a custom CNN backbone unrelated to the ResNet-based LibMTL ecosystem. Integrating this setting into LibMTL would require non-trivial architectural design decisions for every baseline, which would make comparisons less standardized.
> - This implementation overhead is precisely the limitation our paper discusses: many MTL methods require substantial architectural modifications to adapt to new backbones. In contrast, TSσBN is a drop-in replacement with no additional routing, attention, or branches, and thus applies cleanly in all settings.
>
> ### 2. **Additional Baselines**
> - MTAN on CelebA is already included in Table 1.
> - DSBN is effectively equivalent to TSBN: both introduce per-task (or per-domain) BN affine parameters while sharing convolutions. We agree that the naming could be clearer and will make this connection explicit.
> - TaskNorm is not applicable here: it is designed for meta-learning and addresses mismatched feature statistics (μ, σ) arising from domain differences. Our work targets single-domain MTL, where all tasks share the same input and does not suffer from mismatched statistics. The novelty lies in task-specific γ-modulation for capacity allocation and interference mitigation rather than statistical correction.
>
> ### 3. **Novelty**
>
> We would appreciate more specific guidance on which aspects appear insufficiently novel. To clarify:
>
> - Prior work on conditional BN focuses on statistical correction (μ, σ) due to domain shift, whereas our contribution centers on the **post-normalization transform**, showing that task-specific γ-gating alone is a sufficient mechanism for MTL.
> - No prior MTL work uses **σ-gated BN variations or discriminative learning rates** to control capacity allocation or task interference.
> - Our method yields a **novel built-in interpretability framework** (filter importance, capacity decomposition, stable task hierarchies), which to our knowledge, is the first unified perspective derived directly from normalization parameters.
> - TSσBN applies to **both CNNs and Transformers** with the same formulation, whereas many existing MTL methods are architecture-specific.
> - TSσBN consistently **matches or surpasses state-of-the-art** methods across seven benchmarks while using far fewer parameters. We have also added comparisons against recent parameter-efficient adapters (e.g. MTLoRA, Compactor, VPT) and TSσBN outperforms these as well.
>
> ### 4. **FLOP Clarification**
> - FLOPs in Table 2 are full inference FLOPs including task heads.
> - TSBN and TSσBN have nearly identical FLOPs since σBN only adds a sigmoid on γ (γ·x + β vs. σ(γ)·x).
> - This overhead is negligible compared to methods that introduce additional convolutions, routing modules, or experts, which substantially increase FLOPs.
>
> We hope these clarifications address the reviewer’s concerns. Given the simplicity, generality, and strong empirical performance of our method, together with the added comparisons and explanations, we respectfully ask the reviewer to consider raising their score. We would be happy to provide any additional details or discussion that may assist in the evaluation.

---

### Author Response · Authors · 2025-11-26
**Author Follow-Up**

Dear Reviewers,

Thank you for your feedback. We have addressed all comments raised in the reviews and updated the manuscript to include additional experiments, revised comparisons, and clarifications where needed.

If there are remaining questions or points that require further explanation, we are available to provide them. Otherwise, we kindly ask you to review your scores in light of the rebuttal and revised paper.

Thank you for your time.

---

### Author Response · Authors · 2025-12-03
**Author Final Remarks**

We thank the reviewers and ACs for their time. We recognize that reviewing resources, especially under these circumstances, are limited. Hence, we provide a concise summary of the reviews, key misunderstandings, and how our rebuttal and revision addressed them.

---

### Strengths

Reviewers consistently acknowledged three core strengths of our contribution:

1. **Simplicity and practicality**: A minimal, drop-in mechanism that mitigates interference in MTL and is easy to integrate.
2. **Robustness**: Stable performance across different hyper-parameters and loss scales, distinguishing it from generic task-specific BN.
3. **Interpretability**: The task–filter importance matrix and resulting ablations are valuable tools for understanding and diagnosing MTL models.


---

### Weaknesses and Rebuttal


**1. Novelty**

It was assumed that our contribution is simply task-specific BN for MTL, a technique already known in other contexts. We clarified:

- **Different motivation and focus**: Prior BN work targets domain shift (in various forms) and modifies BN’s normalization. Our work is in single-domain MTL where normalization is a non-issue; we focus instead on the **post-normalization** transformation as a sufficient mechanism for capacity allocation and interference mitigation.

- **Different implementation**: We use **σBN**, a bounded gating formulation explicitly designed for feature importance, trained with **high discriminative learning rates**, which is a clear departure from standard per-task BN.

- **Additional utility**: This parametrization yields a **novel unified interpretability framework** (capacity decomposition, task similarity, feature specialization) which to our knowledge has no counterpart in MTL literature.

- **Clearer positioning**: Although the referenced works were discussed in the original draft, we expanded the relevant sections and comparisons to avoid any further confusion.


**2. Benchmarks and Recency**

- **Breadth of evaluation**: While we agree that more evaluation is better, our submission already exceeds typical MTL standards:
5 architectures, 4 datasets and **7 experimental setups**, which is far more than the common “1 backbone + 1–2 datasets.”. Furthermore, our method is widely applicable to any domain or architecture or that uses normalization layers.

- **Modernity of baselines**: Beyond existing SOTA MoE comparisons we included a full MTLoRA-style PEFT benchmark, evaluating against SOTA adapter based methods. On both **ViT and Swin** backbones TSσBN achieves superior performance with fewer parameters and far lower complexity.

- **Optimization-based baselines**: Upon request we also added comparisons to MTL optimization based methods. Although orthogonal, our method performed best.

All other comments regarding FLOPs, hyper-parameters, terminology were fully addressed.

---

### Closing Note

Our rebuttal directly resolved the reviewers’ concerns and clarified several misunderstandings in the initial reviews. The revision further strengthens our paper with new experiments also showing state-of-the-art performance against adapter based MTL methods, as well as clearer articulation of novelty and positioning.

---

### Meta-Review · Area_Chair_mAPD · 2025-12-29

**Summary:**

The concerns are centered around

1. **Experimental Gaps & Comparisons** (two reviewers shared the concern): Missing baselines such as GradNorm or CAGrad; limited datasets, lacking results on CelebA and at least one large-scale, more challenging benchmark such as Taskonomy; missing comparisons against methods that modify gradients rather than only model structure.

1. **Novelty** (two reviewers shared the concern): Ideas appear similar to per-task normalization approaches (e.g., Task-Specific BN), but the work does not sufficiently discuss or differentiate from prior art; the technical contribution is viewed as incremental and below the high bar for ICLR; insufficient theoretical justification.

1. **Sensitivity Study**: More extensive analysis is needed on the hyperparameters $\gamma, \tau$ to demonstrate stability and robustness.

1. **Outdated Backbone**: The empirical evaluation relies on older architectures like ResNet-50 and ViT-S, missing the SoTA backbones such as ConvNeXt and DETR

**Reviewer Concerns:**

The authors responded by:

1. **Experimental Gaps & Comparisons**. They argue that the selected benchmarks align with standard MTL evaluations in recent SOTA work. They mention that using larger datasets is difficult due to compute cost. They also added a comparison to an optimization-based baseline in the rebuttal.

1. **Novelty**. The authors clarify the differences from prior work, including distinctions in motivation and technical design.

1. **Sensitivity Study**. The authors explain their hyperparameter setup and argue that their approach requires a lower tuning burden compared to baseline methods.

1. **Outdated Backbone**. The authors reaffirm that ViT-S and Swin-T are widely used transformer-based backbones in MTL research. They further state that MTL benchmarks for ConvNeXt or DETR are not yet established, but their method is applicable to any architecture with normalization layers.

The reviewers’ concerns seem to be only partially addressed, especially the first point regarding the empirical verification study. While the authors may be correct that their setup aligns with recent research practice, it would strengthen the submission to include results on a large-scale, more challenging benchmark together with SoTA architectures, as explicitly requested by two reviewers. This would better substantiate their claims on effectiveness and broad applicability, and further reinforce the novelty through stronger empirical evidence.

**Reviewer Scores:**

`Reviewer wFix` (initial 4) might increase to 6 due to the clarifications provided, but not higher because of the remaining concerns on novelty.

`Reviewer gf9H` (initial 2) is likely to increase the score to 4 but not higher, given unresolved novelty concerns and the missing backbone/dataset comparisons explicitly requested.

`Reviewer BXZU` (initial 6) may retain the same score or slightly increase the rating due to the effective clarifications and added experiments.

---

### Decision · Program_Chairs · 2026-01-26

Reject